# Deep Bi-LSTM Networks for Sequential Recommendation

**DOI:** 10.3390/e22080870

**Published:** 2020-08-07

**Authors:** Chuanchuan Zhao, Jinguo You, Xinxian Wen, Xiaowu Li

**Affiliations:** 1Faculty of Information Engineering and Automation, Kunming University of Science and Technology, Kunming 650504, China; zhaochuanchuan@stu.kust.edu.cn (C.Z.); wenxinxian@stu.kust.edu.cn (X.W.); lxwlxw66@kmust.edu.cn (X.L.); 2Computer Technology Application Key Lab of Yunnan Province, Kunming 650504, China

**Keywords:** recommendation systems, interactive sequence, class label, deep bidirectional LSTM, self-attention, item similarity

## Abstract

Recent years have seen a surge in approaches that combine deep learning and recommendation systems to capture user preference or item interaction evolution over time. However, the most related work only consider the sequential similarity between the items and neglects the item content feature information and the impact difference of interacted items on the next items. This paper introduces the deep bidirectional long short-term memory (LSTM) and self-attention mechanism into the sequential recommender while fusing the information of item sequences and contents. Specifically, we deal with the issues in a three-pronged attack: the improved item embedding, weight update, and the deep bidirectional LSTM preference learning. First, the user-item sequences are embedded into a low-dimensional item vector space representation via Item2vec, and the class label vectors are concatenated for each embedded item vector. Second, the embedded item vectors learn different impact weights of each item to achieve item awareness via self-attention mechanism; the embedded item vectors and corresponding weights are then fed into the bidirectional LSTM model to learn the user preference vectors. Finally, the top similar items in the preference vector space are evaluated to generate the recommendation list for users. By conducting comprehensive experiments, we demonstrate that our model outperforms the traditional recommendation algorithms on Recall@20 and Mean Reciprocal Rank (MRR@20).

## 1. Introduction

In the age of the Internet, users are used to acquiring the items or information on their demand from the Internet. However, with the increasing number of online users and the explosion of information on the Internet, users are facing the challenges of obtaining the information they really need. As a result, the recommendation systems emerge with the purpose of digging out items that are of interest to each user from a huge collection of items, and presenting them to users.

Generally, the recommendation list is obtained based on the user preferences, the item features, and other auxiliary information. The recommender systems are roughly divided into three types: content-based recommendation, collaborative filtering, and hybrid recommendation [1].

Collaborative filtering (CF) [2] and content-based recommendation [3] are traditional methods of recommender systems. The user-based collaborative filtering (UCF) tends to recommend to target users the items that are highly scored by the other users who are similar to the target users. Lots of literature calculated the similarity among users or items by using a CF from user interaction data (i.e., the scoring matrix). However, the traditional recommendation methods are mainly focused on recommending similar items to users, without considering the content relevance between an item and other context items in the item sequence.

Recent years have seen a surge in approaches that combine deep learning and recommendation systems [4]. Deep learning not only well trains some non-linear or non-trivial implicit relationships between users and items, but also efficiently encodes the underlying complex abstract meanings of the network into higher-level data representations. In addition, deep learning can learn the intricate relationship of data itself from the rich data resources that are easily available. Due to these high-performance features of deep learning, its applications in recommender systems have gradually become more widespread in recent years [5,6,7,8,9,10,11,12], but there are also corresponding problems. The sequence recommendation methods among them only adopt the sequential similarity between items, and neglected the item content information which usually is the indispensable complement. Moreover, in the real-life application scenarios, users often focus on certain items while ignoring others based on the interacted items which have different impact on the next item choice.

To address the above issues, this paper proposes a recommendation model by fusing the item sequences and contents (FISC). The model introduces the user preferences and deep recurrent neural networks, such as the long short-term memory (LSTM) and self-attention, where the sequence that consists of the user interacted items and class labels are fed into a recurrent neural network to improve the precision of recommending a system. At the same time, the different influence weights $a_{i}$ learned by the self-attention mechanism are also input into the deep bidirectional LSTM. After multiple training, the last layer of the model outputs the user preference vectors. Finally, all the items are ranked by the similarity between the preference vectors and the embedded vectors, and the Top-k items are selected as the recommendation list of the users.

Our main contributions are threefold.

We add item content information into Item2vec to represent items more abundantly and comprehensively. First we use Item2vec to learn the initial embedded vector representation of items, then vectorize their content attribute information, and finally concatenate the two vectors as the final representation of items.In order to learn the different impact of each item in the interaction sequence on the candidate items, we introduced the self-attention mechanism into the model. We leverage Bi-LSTM to learn the user preference representation bidirectionally from the user item sequence for more context information, and enhance the shallow Bi-LSTM to a deep Bi-LSTM to learn the user deeper preference representation.We made a comparison of the experimental results of the self-attention mechanism and deep Bi-LSTM on the recommendation effect. The experimental results on the real-world dataset have demonstrated the effectiveness of our recommendation model.

The remainder of this paper is structured as follows. In Section 2, we propose the recommendation model, which takes into account the content information of items and the relationship between the context items, based on the deep Bi-LSTM and self-attention. Section 3 presents an experimental analysis based on the existing public dataset. Section 4 discusses some existing related work in the recommender system. Finally, our work is summarized in the last section.

## 2. Sequential Recommendation Model

The deep recurrent neural networks perform well in natural language processing (NLP). As the LSTM in recurrent neural networks has recently been successfully applied in sequence prediction problems, this paper leverages the deep Bi-LSTM and self-attention to learn user preference information on a deeper level.

Our sequential recommendation model includes three main phases: item embedding, weight update and user preference learning. Figure 1 shows the our model framework.

### 2.1. Item Embedding

Given all user interaction item sets $S=\left(S_{1},S_{2},S_{3},\ldots ,S_{m}\right)$, where $S_{i}$ represents item sequence $S_{i}=\left\{I_{1},I_{2},I_{3},\cdots ,I_{n}\right\}$ of the user *i*, the purpose of item embedding is to generate low-dimensional item vectors for each item. This paper only selects sequences that show feedback on user preferences. For example, if a user scores a low-interest item, the user is not interested in the item. Traditional item embedding often only considers the second-order correlation between items and does not consider the relationship between item attributes and content. We embed the class labels of an item into the item vector, which can better calculate the relevance of the item and learn the user preferences.

Item2vec [13] is one of the important extensions of Skip-gram and negative sampling [14] for item embedding for item-based collaborative filtering recommendations. To introduce the class label into the model, this paper has made some improvements to Item2vec: the class label of an item is one-hot encoded to obtain a vector, and this vector is connected with the embedding vector learned by Item2vec to obtain the final embedded representation vector of the item. Similar to Word2vec, this paper treats each item as a word. The sequence of user-interacted items is treated as a sentence, and each item is embedding into a vector of fixed dimensions. Each user has an item sequence of interactions that is different from each other. Finally, by embedding each user’s item sequence of interactions to obtain a fixed-dimensional item vector, the closer the vectors are, the more similar the vectors are in the embedding space. The process of item embedding is shown on the top of Figure 1.

Given a user’s item sequence of interactions, the Skip-gram goal is to maximize the following objective functions:(1)$$\frac{1}{M}\sum_{x=1}^{M}\sum_{y\neq i}^{M}\log p\left(I_{y}|I_{x}\right),$$
where M is the length of the item sequence of interactions, $p\left(I_{y}|I_{x}\right)$ is the SoftMax function:(2)$$p\left(I_{y}|I_{x}\right)=\sigma \left(I_{x}^{T}I_{y}\right)\prod_{k}^{N}\sigma \left(-I_{x}^{T}I_{j}\right),$$
where σ(∗) is a commonly used Sigmoid activation function and *N* is the number of negative samples in the positive sample. By item embedding, we get the item sequence $S_{i}=\left\{V_{1},V_{2},V_{3},\ldots ,V_{n}\right\}$.

### 2.2. Weight Update

The attention mechanisms are recently widely employed as a powerful tool in sequence data scenarios such as machine translation, speech recognition, and part-of-speech tagging. The attention mechanism can be used alone or mixed with other models. It uses the automatic weighted transformation of data to connect two different parts to highlight key data, so that the entire model can reach better performance. The attention mechanism is similar to the principle that the human brain observes certain things. For example, when the people observe a certain painting for describing the content of the painting, people first observe the words on the painting, and then make a purposeful observation for the part of the picture that represents the theme from their judgment. When people describe the painting, they often first describe the content that is most relevant to the painting, and then describe other aspects; the self-attention mechanism also is a mechanism by assigning sufficient attention to the key information and highlighting locally important information.

In real life, user preferences are not static. When users focus on certain products, they may potentially ignore other products. For example, as shown in Figure 2, the sequence of items that the target user interacts with is “Movie A, Movie B, Movie C, Movie D, Movie E”. We want to predict what’s his next favorite movie. If the most recent interactions (movie C, movie D, movie E) are regarded as the context and are given higher priority, it is likely that the items recommended to the user are “comedy” movies, such as “Movie F”. However, the actual interaction record shows that the user chooses “Movie G” as the next item choice, because the choice of “Movie G” may depend on the first two items (Movie A and Movie B) that the user actually interacts with. This indicates that a good recommender system should pay much attention to those items (movie A and movie B) that are more related to the target item (movie G), rather than the newly added but less relevant items like “Movie C, Movie D, Movie E”.

Therefore, this paper further proposed an item-aware weight update model based on the principle of self-attention mechanism. This model uses the self-attention mechanism to model the internal relationship between user-interactive items when learning the user’s potential preference representation, so that the user preference representation is more effective, as shown in the middle of Figure 1.

The weight update introduces self-attention mechanism which we elaborate here. First, for a fixed target item, it traverses the state of all encoders to compare the state of the target item with that of the source item (i.e., the relationship between the target item and the source item), so as to generate a score for each state in the encoder. Second, the SoftMax function is used to normalize all scores to generate a probability distribution given the target item state. Finally, we obtain the item weight from the distribution.

In essence, the item feature representation is mapped from *d*-dimensional space to *z*-dimensional space. The relationship mapping is shown as follows:(3)$$l_{z}=f_{Relu}\left(WI_{d}+b\right)$$
(4)$$A=Softmax\left(I_{z}W{\left(I_{d}\right)}^{T}\right)$$
(5)$$I_{z}=AI_{z},$$
where *W* is a weight matrix, *b* is bias and $l_{z}$ is the feature representation vector after item feature embedding. Equation (Equation 3) indicates that the user interaction item features in the *d*-dimensional space are mapped to the *z*-dimensional space. Equation (Equation 4) calculates the contribution weight of all user interaction items in the *d*-dimensional space to each user interaction item in the *z*-dimensional space. The model automatically adjusts the weight matrix *W* through the loss function during model training, and the matrix *A* normalized by the Softmax function. The items in the *z*-dimensional space are weighted. After weighting, the feature representation of each item in the *z*-dimensional space is jointly represented by itself and all the items associated with it. The final output $I_{z}$ is the characteristics of the item after weighted by the self-attention mechanism. By weight update, we get the item different weight $a_{i}$.

### 2.3. User Preference Learning

RNN (Recurrent Neural Networks) play a vital role in predicting the next target of the sequence. Inspired by the literature [15,16], the user item sequence of interactions is treated as a sentence, and each item can as a word. We use deep recurrent neural networks to learn the relevance of each item in the item sequence of interactions to the adjacent item. This paper is based on deep bidirectional LSTM (i.e., deep Bi-LSTM), as shown in Figure 1, enabling the model to better utilize forward and backward contexts representation. And deep recurrent neural networks also can better extract user characteristics.

Figure 1 shows the preference modeling, which has a double hidden layer, and information of each upper layer in the structure is provided by its lower layer. As the picture shows, in the network structure, the previous time step generates a set of parameters and passes the set of parameters to the inter-neurons in the same Bi-LSTM layer at a later time step *t*. At the same time, the inter-neuron needs to receive two sets of related parameters from the previous layer of the Bi-LSTM hidden layer in the time step *t*; the input sequence of each hidden layer in the model starts from two directions: from left to right, from right to left.

The relational of the deep Bi-LSTM structure is denoted in Equations (Equation 6) and (Equation 7). At the same time step *t*, each output of the layer Bi-LSTM of r−1 layer serves as an input to each intermediate neuron of the *r* layer. At each time step in the training model, the result is produced by the hidden layer propagation via connecting all the input parameters. The last hidden layer produces the final output *P* (Equation (Equation 8)).
(6)$${\vec{h}}_{t}^{\left(r\right)}=f\left({\vec{A}}^{\left(r\right)}{\vec{h}}_{t}^{\left(r-1\right)}+{\vec{B}}^{\left(r\right)}{\vec{h}}_{t-1}^{\left(r\right)}+{\vec{z}}^{\left(r\right)}\right)$$
(7)$${\overset{\leftarrow }{h}}_{t}^{\left(r\right)}=f\left({\overset{\leftarrow }{A}}^{\left(r\right)}{\overset{\leftarrow }{h}}_{t}^{\left(r-1\right)}+{\overset{\leftarrow }{B}}^{\left(r\right)}{\overset{\leftarrow }{h}}_{t+1}^{\left(r\right)}+{\overset{\leftarrow }{z}}^{\left(r\right)}\right)$$
(8)$$\vec{P}=\mathrm{concat}\left({\vec{h}}_{t}^{\left(r\right)},{\overset{\leftarrow }{h}}_{t}^{\left(r\right)}\right),$$
where ${\vec{A}}^{\left(r\right)}$, ${\vec{B}}^{\left(r\right)}$ and ${\vec{z}}^{\left(r\right)}$ are weight matrix and offset vector generated in forward propagation at the *r* layer of the model; ${\overset{\leftarrow }{A}}^{\left(r\right)}$,${\overset{\leftarrow }{B}}^{\left(r\right)}$ and ${\overset{\leftarrow }{z}}^{\left(r\right)}$ are weight matrix and offset vector generated in backward propagation at the *r* layer of the model; $\vec{P}$ is output vector; ${\vec{h}}_{t}^{\left(r\right)}$ and ${\overset{\leftarrow }{h}}_{t}^{\left(r\right)}$ are respectively the intermediate representation of the past and the future is used to discriminate the input vector.

The *r* layer propagation of the model is based on the hidden state ${\vec{h}}_{t}^{\left(r-1\right)}$ of the current moment of the previous layer and the hidden state ${\vec{h}}_{t-1}^{\left(r\right)}$ of the previous moment of the current layer to calculate the forward direction of the hidden state ${\vec{h}}_{t}^{\left(r\right)}$ of the current layer at the current moment; In contrast, backward propagation requires the hidden state ${\overset{\leftarrow }{h}}_{t}^{\left(r-1\right)}$ of the current layer and the future state ${\overset{\leftarrow }{h}}_{t+1}^{\left(r\right)}$ of the current layer to calculate the hidden state ${\overset{\leftarrow }{h}}_{t}^{\left(r\right)}$ of the update reverse. Thus, each hidden representation can be computed using a series calculation function $\mathrm{concat}\left({\vec{h}}_{t}^{\left(r\right)},{\overset{\leftarrow }{h}}_{t}^{\left(r\right)}\right)$ that concatenates the forward and backward hidden representations. The last layer of the hidden layer outputs the preference vector through the fully connected layer.

The embedded vector of each item is used as the input of the model. In the model training process, the mean square error (MSE) and Adagrad are used to learn the optimization model, so that the model can well learn the preferences of each user and better understand and represent the user long-term stable preference.

The MSE equation is shown as follows:(9)$$MSE=\frac{1}{n}\sum_{i=1}^{m}{\left(y_{i}-{\hat{y}}_{i}\right)}^{2},$$
where $y_{i}$ is the actual user interaction item in the test set, $\hat{y_{i}}$ is the predicted item, and $w_{i}$ > 0 is the item weight. The more similar the predicted item and the actual interaction item are, the better the model performs, meaning the more accurate its prediction is.

### 2.4. Algorithms

The entire process of our recommender system includes Algorithm 1 Item Embedding and Algorithm 2 User Preference Learning as follows.
**Algorithm 1** Item Embedding**Input:**  $S=\left(S_{1},S_{2},S_{3},\ldots ,S_{m}\right)$ - All user item sequences;  $\vec{L}$ - Class label vector.**Output:**  ${\vec{V}}_{j}$ - A vector representation of per items in a low-dimensional space;  $U_{i}$ - matrix corresponding to user *i*’s item sequence.1: **for** each i∈[0,m−1]
**do**2:  Feed $S_{i}$ of user *i* into Item2vec; $\frac{1}{M}\sum_{x=1}^{M}\sum_{y\neq i}^{M}\log p\left(I_{y}|I_{x}\right);$3: **end for**4: **for** each j∈[0,n−1]
**do**5:  ${\vec{V}}_{j}={\vec{V}}_{j}+{\vec{L}}_{j};$6: **end for**7: **return**    $S_{i}=\left\{V_{1},V_{2},V_{3},\ldots ,V_{n}\right\}$

In Algorithm 1, the user item sequences and the content of the items are fed into Item2vec to train the item’s embedding vectors.

First, we extract the item set I and the item’s label set L from the user rating data.

Then, we convert the item set and the label set into a one-by-one item sequence $S_{i}$ and a one-hot encoded label vector $\vec{L}$, respectively.

Finally, the item sequence and label vectors are learned as the embedded vector representation ${\vec{V}}_{i}$ of items in low-dimensional space by Item2vec model (Line 1–5).

Algorithm 2 is to feed the item’s embedding vectors corresponding to the user item sequence into deep Bi-LSTM, which generated the user preference vector by optimization of the model.

First, we use deep Bi-LSTM to learn user preferences vector ${\vec{P}}_{i}$, and add multiple hidden layers to enhance the model’s expressive ability (Line 1–11).

Then, we calculate the similarity ${\mathrm{sim}}_{U_{i},I_{j}}$ between the preference vector ${\vec{P}}_{i}$ of the target user $U_{i}$ and the vector ${\vec{V}}_{j}$ of each item learned in the low-dimensional space: ${\mathrm{sim}}_{U_{i},I_{j}}={\vec{P}}_{i}\cdot {\vec{V}}_{j}$ (Line 12–16).

Last, we filter out the item set $I^{\prime }$ that the target user does not interact with, and sort items according to the similarity ${\mathrm{sim}}_{U_{i},I_{j}}\left(I_{j}\in I^{\prime }\right)$. Top-*k* items are recommend to the target users (Line 17).
**Algorithm 2** Weight Update and User Preference Learning**Input:**  $S_{i}=\left\{V_{1},V_{2},V_{3},\ldots ,V_{n}\right\}$ - The first j−1 item set of all item sequences;  $a_{i}$ - Impact weight of item *i* on the next item selection;  length - The length of item sequence.**Output:**  ${\vec{P}}_{i}$ - The preference vector of the target user $U_{i}$;  A recommendation Top-*k* list.1: **for** each j∈[0,n−1]
**do**2:  **if**
j< length-1 **then**3:    $V_{j}$ and $a_{i}$ input to deep Bi-LSTM;4:  **else if**
*j* = length-1 **then**5:    $V_{j}$ as a target item of deep Bi-LSTM;6:  **else**7:    break;8:  **end if**9: **end for**10: MSE optimization, parameter update;11: **return**
${\vec{P}}_{i}$;12: **for** each i∈[0,m−1]
**do**13:  **for** each j∈[0,n−1]
**do**14:   
${\mathrm{sim}}_{U_{i};I_{j}}={\vec{P}}_{i}\cdot {\vec{V}}_{j}$;15:  **end for**16:  Sort items according to ${\mathrm{sim}}_{U_{i},I_{j}}\left(I_{j}\in I^{\prime }\right)$;17: **end for**18: 
**return** A Top-*k* recommendation list.

Note that a item sequence consists of items that the user $U_{i}$ has interacted with, which is denoted by $\vec{S_{i}}\;=\left\{V_{1},V_{2},\cdots ,V_{n}\right\}$, and the ${\vec{P}}_{i}$ of the model training output represents the user preference vector.

## 3. Experimental Evaluation

To evaluate our proposed recommendation model, this paper conducted the following experiments by comparing with the state of the art of recommender methods.

### 3.1. Datasets

The dataset is the MovieLens 10m dataset, which is a classic dataset to calibrate various algorithms in recommender systems. It contains 710,054 ratings and 20 class label information for 71,567 users and 10,681 movies. The users selected in the data set are the users who have scored at least 20 movies. The user’s rating data for the movie is contained in the ratings file, where each line represents a user’s rating for a movie. The ratings file is sorted according to the order of the *userId*, and then sorted in the order of the *movieId*. The movies file contains content information about the movie, and each line represents a movie. Among them, *movieId* is the real movie Id with which a movie is uniquely identified and *genres* is the category information of the movie. Note that a movie may belong to multiple genres or categories.

Table 1 and Table 2 are samples of the item’s sequence and item’s contents, respectively.

The data set is divided into two parts—the training set and the test set. The 90% of the data is randomly used as the training set to complete the training of the whole algorithm, and the remaining 10% of the data is used as the test set to measure the actual performance of the model. When training the model, the items with less than 3 training users in the training set are filtered from the item sequences, which makes the model perform better and more appropriate to the user preference.

### 3.2. Baseline Algorithms

To verify the validity of the proposed model, the following algorithms are given as comparisons:

Item-based k-NN: An item-based nearest neighbor recommendation. It is a collaborative filtering method that determines the target user’s rating of the item, finds other items that are similar to the item, and infers his rating of the item based on the target user’s rating of a similar item. In Reference [17], the non-personalized variants of the Item-based k-NN achieved good recommendation results in the sequence prediction task.

Exp. Dec. Item-based k-NN [18]—An exponential decay component is added to the Item-based k-NN to punish items that were consumed early.

Matrix Factorization (MF) [19]: A typical model-based collaborative filtering recommendation algorithm that uses the rating information between users and items to predict the target user’s rating of the item. However, MF only uses scoring data. Due to the scoring matrix is a very sparse matrix, MF has serious data sparsity problems, and it is a shallow model that cannot learn further features between users and items.

Seq. Matrix Factorization [20,21] is a method of adding a sequence based on MF and expanding to a sequence MF method.

Standard GRU [22]: Some recommendation methods take advantage of information about user interaction behavior. For example, Reference [22] uses the standard GRU to embed the user interaction items and its corresponding actions into a model to learn the interaction model of the user, and finally outputs the next possible interaction action of the user.

### 3.3. Evaluation Criterion

Two ranking-based evaluation indicators are used to evaluate the quality of the next recommendation:

Recall @*k*: It is the main evaluation indicator. Defined as the next item of the target user’s actual item sequence appears in the Top-*k* recommendation list, which is the proportion of cases with the required items in the Top-*k* items in all test cases [23].
(10)$$Recall@k=\frac{Num}{Num_{G}},$$
where Num is the number of related items in the recommended top-*k* items, and $Num_{G}$ is the number of all related items.

MRR @*k*: Another evaluation criterion is the Mean Reciprocal Rank, the average of the reciprocal ranking of the next item of the target user’s actual item sequence in the recommendation list [24]. If it is higher *k*, the level is set zero.
(11)$$MRR@k=\frac{1}{\left|Q\right|}\sum_{x=1}^{\left|Q\right|}\frac{1}{rank_{x}},$$
where |Q| is the number of items that the test centralized user interacts, and $rank_{x}$ is the item where the user interacts in the *x*-th position of the recommendation list.

In short, the higher the value of these two evaluation indicators are, the better the recommendation results of our model is. (In this paper, we set *k* = 20).

### 3.4. Experimental Results

#### 3.4.1. Parameter Configuration

In order to ensure the accuracy of the algorithm results, this article uses the parameter configuration corresponding to the original text, the parameters of the three comparison algorithms are configured as follows:

Item-based k-NN nearest neighbors is set to 100; Exp. Dec. Item-based k-NN has a decay constant set to 1; The number of the potential factor of Matrix Factorization (MF) and the number of iterations are set to 20 and 30, respectively; The window size of Seq. Matrix Factorization is set to 2; The batch and iteration times of standard GRU are set to 1000 and 10, respectively.

#### 3.4.2. Performance Comparison

Table 3 shows the performance comparison in terms of the two evaluation indicators Recall@20 and MRR@20.

The proposed model produces better results than other comparison methods. Especially on the Recall@20 indicator, after adding the item content information, the recommendation performance is significantly improved compared to the Standard GRU, and at the same time, the MRR@20 indicator has also been improved.

The performance of the recommendation system has been improved by adding time and sequence factors from the traditional method. It can be seen from the Item-based k-NN and Exp. Dec. Item-based k-NN that the calculation steps are the same, and the time factor can improve the performance. Prove that using sequence recommendations can better mine users’ interest preferences. From the comparison of Matrix Factorization (MF) and Seq. Matrix Factorization, it is found that by adding user item sequence factors, user preferences can be better modeled, and the hidden correlation between items can be obtained from the sequence.

The proposed model is significantly improved compared with the standard sequence recommendation GRU. The key point is that the item embedding process combines the content information of the item and the design of the neural network, which not only considers the order similarity of the item but also considers the item’s content similarity. Further, a deeper neural network can better learn user preferences.

#### 3.4.3. Impact of Embedded Dimensions

The dimensions of the item’s embedded representation also affect our model’s performance.

Figure 3 shows the effect of the size of the embedded dimension on the performance of the proposed model in the item embedding process. From the experimental results in the Figure 3, it can be seen that when the embedding dimension is 8 dimensions, the embedded representation of the item does not reach the best, and the recommendation performance of the trained model is relatively low. Meanwhile, the main evaluation index Recall@20 is worse than the 16-dimensional result. When the embedding dimensions are 16 and 32, the two index values corresponding to the two dimensions are almost equal, and the performance of the model is improved compared to the 8-dimensional index value, but this does not mean that the higher the dimension is, the better the recommendation performance of the model is. When the dimension is 20, the performance of the model reaches the highest value relative to other dimensions on the two indicators, and the recommendation performance is relatively good.

#### 3.4.4. Impact of Deep Bi-LSTM

In order to further evaluate the efficiency of the proposed deep neural network, we compared the experimental results of single-layer bidirectional LSTM (Bi-LSTM) and deep bidirectional LSTM shown in Table 4. It can be seen that the proposed deep model is much improved compared to the single-layer model. From the two evaluation indicators Recall@20 and MRR@20, the recommendation performance results for deep Bi-LSTM are 1.59% and 2.91% higher than Bi-LSTM, respectively. And the key is that deep neural networks can learn a deeper representation of preferences between item sequences.

#### 3.4.5. Impact of Self-Attention

According to the experimental results as shown in Table 5, the introduction of the self-attention mechanism makes our recommendation model effectively improve the Recall@20 and MRR@20 indicators by 8.26% and 7.34%. Because the self-attention mechanism can learn the weight that is the impact of items which are from the item sequences on candidate item (the next item that the user is about to interact with), and assign various weights to these items to achieve item awareness. The model can more accurately express user preferences.

Our model performs much better than other traditional advanced algorithms. The key lies in the combination of deep bidirectional LSTM and self-attention neural network. Deep learning can better mine the potential characteristics of users, deeply model the correlation between items, and improve the performance of recommender systems.

## 4. Related Work

Reference [6] employed a bidirectional deep neural network with two hidden layers for extracting reviews at each time, where a hidden layer is used for forward propagation (from left to right) and another layer for back propagation (from right to left). Furthermore, the efficiency of the model is proved by their experiments. To be able to well maintain these two hidden layers at all times, the network’s weight and offset parameters consume twice as much memory space. Finally, the final classification result is generated by combining the result scores produced by the two-layer RNN hidden layer. Equations (Equation 12) and (Equation 13) are used to calculate the hidden layer representation of the bidirectional RNN. The only difference between the two hidden layers is that they are recursively different through the corpus. Equation (Equation 14) is a comprehensive representation of the hidden learning in both directions, thereby predicting the possible classification relationship of the next word.
(12)$${\vec{h}}_{t}=f\left({\vec{A}}_{x_{t}}+\vec{B}{\vec{h}}_{t-1}+\vec{z}\right)$$
(13)$${\overset{\leftarrow }{h}}_{t}=f\left({\overset{\leftarrow }{A}}_{x_{t}}+\overset{\leftarrow }{B}{\overset{\leftarrow }{h}}_{t-1}+\overset{\leftarrow }{z}\right)$$
(14)$${\hat{y}}_{t}=g\left(W{\vec{h}}_{t}+c\right)=g\left(W\left[{\vec{h}}_{t};{\overset{\leftarrow }{h}}_{t}\right]+c\right),$$
where $\vec{A}$, $\vec{B}$ and $\vec{z}$ are weight matrix and offset vector generated in forward propagation; $\overset{\leftarrow }{A}$, $\overset{\leftarrow }{B}$ and $\overset{\leftarrow }{z}$ are weight matrix and offset vector generated in backward propagation; *U* is output matrix; ${\vec{h}}_{t}$ and ${\overset{\leftarrow }{h}}_{t}$ are respectively the intermediate representation of the past and the future is used to discriminate the input vector, and *c* is the output deviation.

In Reference [7], Feng et al. proposed an MN-HDRM recommendation model. It mainly uses the neural network framework to fuse two types of neural network models—the recurrent neural network is responsible for modeling dynamic data, and the forward neural network is responsible for global user data modeling. However, this method uses the deep neural network too much, which causes an over-fitting problem on the training learning of the model.

Reference [8] proposed a convolutional LSTM network for predicting precipitation weather. They regard the precipitation prediction in the most recent period as a space-time sequence prediction problem and form a coding prediction by stacking multiple layers of convolutional LSTM layers. The structure of the end-to-end precipitation prediction training model is constructed.

Reference [9] conducted deep research on the time-cycle mode that users sign in terms of time non-uniformity and time continuity. Reference [25] incorporated time-cycle information into a user-based collaborative filtering framework for time-aware POI recommendations.

In the context-aware repetitive neural network [10], the recommender system constructed by RNN also utilizes information containing multiple contexts to process behavior sequence problems. However, there are some problems with RNN when processing sequence data, assuming that the time dependency varies monotonically with the position in the sequence, which means that one element in the sequence is usually more important than the previous element used for prediction. In Reference [11], the authors propose the idea of using RNN and FNN models for a joint recommendation but fail to fully consider the impact of long-term factors.

The most recent work [26,27] uses neural networks to learn preference-preserving binary code embedding on both users and items, which can significantly reduce the storage requirements of the recommendation system.

Interesting areas of recommender systems are building the links between recommendations and cognition or sentiment analysis. Reference [28] suggests that recommender systems should also be designed and evaluated considering cognitive factors to acquiring more intelligence. Reference [29] adjusts the recommendation process and defines a user preference model based on the perception of preferences received by users. Affective computing have great potential as a facility for recommendations to collect feedback on whether the recommended items is in their favors in practice [30].

Although the above methods use deep learning to make recommendations, the content information of the item is not fully utilized, and the above methods neglect the correlation or the impact weights among items. Based on this, this paper proposes a model for fusing item sequences and contents based on the deep bidirectional LSTM model and self-attention, which not only utilizes the user item sequence of interactions but also makes full use of the item’s content information and impact weights to explore deeper relationships between items.

## 5. Conclusions

To improve the performance of recommender systems, this paper proposes the deep Bi-LSTM and self-attention based recommendation model by fusing item sequences and contents. First, the user-item interaction sequence and the item class labels are embedded to obtain the more expressive item sequence vector. Then, the embedded item vectors are fed into self-attention to learn different impact weights $a_{i}$ of each item on candidate item; the sequence vector and $a_{i}$ are fed into the deep Bi-LSTM, which is trained to obtain the preference vector of each user. Finally, the Top-*k* recommendation list is given by calculating the similarity between the preference vector and the item vector. The experimental results demonstrate that the proposed model outperforms the state of the art of the methods compared in this paper on Recall@20 and Mean Reciprocal Rank (MRR@20).

In future work, we intend on adding more item content or user information for embedding to further improve item representation. Another promising direction is that as deep learning is less interpretable, to combine other technologies such as knowledge graph or reinforcement learning with deep learning in the recommendation system will better the user’s experience and increase the desire to purchase. Besides, we will consider training our neural networks in parallel by using Keras and Apache Spark with multi-GPUs to accelerate the training time for big data.

## Figures and Tables

**Figure 1 entropy-22-00870-f001:**
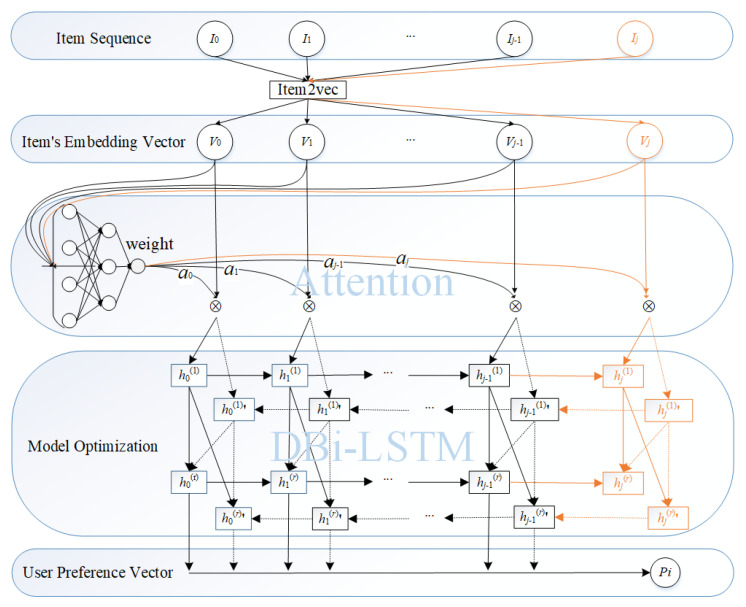
The Sequence Recommendation Model.

**Figure 2 entropy-22-00870-f002:**
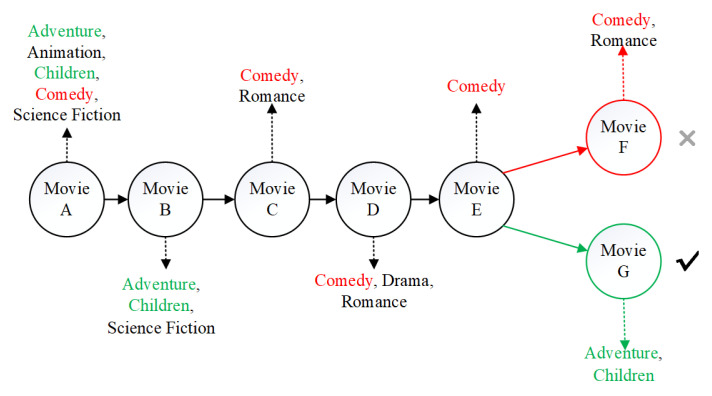
The movie sequence recommendation.

**Figure 3 entropy-22-00870-f003:**
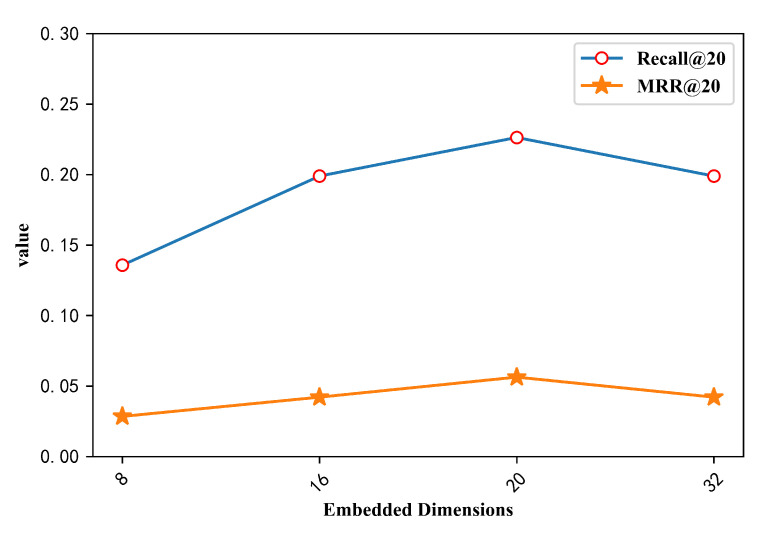
Impact of Embedded Dimensions.

**Table 1 entropy-22-00870-t001:** Item’s Sequence.

UserId	MovieId	Rating	Timestamp
1	122	5	838985046
1	185	5	838983525
1	231	5	838983392
1	292	5	838983421
1	316	5	838983392

**Table 2 entropy-22-00870-t002:** Item’s Contents.

MovieId	Title	Genres
1	Jumanji (1995)	Adventure | Children | Fantasy
2	Grumpier Old Men (1995)	Comedy | Romance
3	Waiting to Exhale (1995)	Comedy | Drama| Romance
4	Father of the Bride Part II (1995)	Comedy
5	Heat (1995)	Action | Crime | Thriller

**Table 3 entropy-22-00870-t003:** Performance Comparison on MovieLens Dataset.

	Recall@20	MRR@20
Item-based k-NN	0.12142	0.03639
Exp. Dec. Item-based k-NN	0.12853	0.04231
Matrix Factorization (MF)	0.07744	0.01192
Seq. Matrix Factorization	0.10730	0.01550
Standard GRU	0.15773	0.04730
**FISC**	**0.22634**	**0.05634**

**Table 4 entropy-22-00870-t004:** Impact of Bidirectional long short term memory (LSTM) with Different Layers.

	Bi-LSTM	Deep Bi-LSTM	Improve
Recall@20	0.22280	**0.22634**	+1.59%
MRR@20	0.05475	**0.05634**	+2.91%

**Table 5 entropy-22-00870-t005:** Impact of Self-Attention.

	no-att	add-att	Improve
Recall@20	0.15040	**0.16283**	+8.26%
MRR@20	0.03186	**0.03402**	+7.34%

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
