# Peer review of "Deep Bi-LSTM Networks for Sequential Recommendation"

_entropy, 2020, doi:10.3390/e22080870_

Round 1

Reviewer 1 Report

This paper discusses Deep Bi-LSTM mechanism for sequential recommender that combines two deep learning and recommendation system approach for reducing the workload to evaluate the recommendation list of the users. Overall the paper is written in a significant way that includes many required entities. However, few queries are to be cleared such as, 1. What if the parallel recommendation model is applied. Would it help to enhance the functionality or could go against the conventional working of recommendation system? 2. The sequential recommendation system is having result output as sequential user preference vector that takes much time as compared to parallel approach so why not adopting the parallel approach? 3. Weight update is taking single input of embedding vectors where as if you further put a classifier then we may apply many weight evaluators and this may goes much faster than the current approach so how this weight is identified in sequenctial order without a classifier? 4. By applying a concat function to the user preferences evaluates the sequential entities in the prefix order, what are the error chances in this evaluation? 5. Related work seems not updated with recent references, we might find some more revelant approach so authors are suggested to add recent references.

Reviewer 2 Report

The manuscript is centered on an interesting topic. Organization of the paper is good and the proposed method is quite novel.

The manuscript, however, does not link well with recent literature on this topic, e.g., check Angulo et al.’s recent special issue on bridging cognitive models and recommender systems. Also, latest trends in binary codes for recommendation systems are missing. Finally, check relevant literature on the related topic of sentiment analysis appeared in top-tier journals, e.g., see the IEEE Intelligent Systems department on "Affective Computing and Sentiment Analysis”.

The manuscript presents some bad English constructions and misuse of articles: a professional language editing service is strongly recommended (e.g., the ones offered by IEEE, Elsevier, and Springer) to sufficiently improve the paper's presentation quality for meeting Entropy’s high standards.

Round 2

Reviewer 1 Report

The authors have addressed the queries and modified the manuscript in a fruitful way. The paper seems a reasonable contribution to the related society.